# High Variability in Sepsis Guidelines in UK: Why Does It Matter?

**DOI:** 10.3390/ijerph17062026

**Published:** 2020-03-19

**Authors:** Alison Bray, Emmanouela Kampouraki, Amanda Winter, Aaron Jesuthasan, Ben Messer, Sara Graziadio

**Affiliations:** 1Northern Medical Physics & Clinical Engineering, Newcastle upon Tyne Hospitals NHS Foundation Trust, Newcastle upon Tyne NE1 4LP, UK; abray3@nhs.net; 2NIHR Newcastle In Vitro Diagnostics Co-operative, Newcastle upon Tyne Hospitals NHS Foundation Trust, Newcastle upon Tyne NE1 4LP, UK; Amanda.Winter@newcastle.ac.uk; 3Translational and Clinical Research Institute, Newcastle University, Newcastle upon Tyne NE2 4HH, UK; Emma.Kampouraki@newcastle.ac.uk (E.K.); aaronjesuthasan@gmail.com (A.J.); 4University College London Hospitals NHS Foundation Trust, London NW1 2BU, UK; 5Department of Anaesthetics, Newcastle upon Tyne Hospitals NHS Foundation Trust, Newcastle upon Tyne NE1 4LP, UK; Ben.Messer@nhs.net

**Keywords:** sepsis, diagnosis, guidelines, POCT, evaluation

## Abstract

It is recommended that developers of Point Of Care Tests (POCTs) assess the care pathway of the patient population of interest in order to understand if the POCT fits within the pathway and has the potential to improve it. If the variation of the pathway across potential hospitals is large, then it is likely that the evaluation of effectiveness is harder and the route towards large-scale takes adoption longer. Evaluating care pathways can be a time-consuming activity when conducted through clinical audits or interviews with healthcare professionals. We have developed a more rapid methodology which extrapolates the care pathway from local hospital guidelines and assesses their variation. Sepsis kills 46,000 people per year in the UK with societal costs of up to £10 billion. Therefore, there is a clinical need for an optimized pathway. By applying our method in this field, we were able to assess the variation in current hospital guidelines for sepsis and infer the potential impact this may have on the evidence development on innovations in this applications. We obtained 15 local sepsis guidelines. Two independent reviewers extracted: use of the national early warning score (NEWS), signs and risk factors informing the decision to prescribe antibiotics, and the number of decisional steps up to this point. Considerable variation was observed in all the variables, which is likely to have an impact on future clinical and economic evaluations and adoption of POCT for the identification of patients with sepsis.

## 1. Introduction

It is recommended that developers of Point Of Care Test (POCT) understand the care pathway of the patient population of interest. The care pathway is the journey of a patient with a specific condition in the healthcare system [1], and its analysis is useful to evaluate whether the new POCT could fit in the current pathway and has the potential to improve it [2,3,4].

Methods to analyze the pathways include review of national guidelines, interviews with stakeholders (e.g., clinicians, laboratory professionals and patients) and audits. These can be time-consuming, especially if the aim is to sample many hospitals. We decided to analyze hospital guidelines as a proxy for the real care pathways. 

Understanding how hospital guidelines are used in practice represents a powerful tool, especially in complex diseases where the definition of the disease is non-trivial. Indeed, guidelines could serve as an operational definition of the disease: when in the pathway initiation of treatment is recommended, then a patient can be considered “diseased”.

When guidelines are not standardized across the country, variations can affect prevalence and, indirectly, clinical and economic evaluation of the POCT (where prevalence of the disease is an essential parameter). Also, adoption of new POCT is harder in a fragmented clinical scenario.

We developed a methodology to evaluate variation in local hospital guidelines and we focused on guidelines for identification of patients with suspected sepsis. This work is relevant to the evaluation of POCTs intended to fit into the sepsis care pathway, for example those based upon procalcitonin (PCT) testing.

Sepsis is a syndrome with a complex definition [5]: it is a dysregulated response to infection that causes system-wide inflammation, which can escalate to multiple organ dysfunction [6]. Because of this complex definition, which is not an operational definition, it is hard to define its prevalence and as a consequence the mortality rate fluctuates substantially between 29% as reported in UK [7] and up to 60% elsewhere [8,9]. It is, therefore, difficult to estimate its clinical and economic impact. Recent estimates report that sepsis kills at least 46,000 people per year in the UK with potential direct costs to the UK National Health System as high as £1.1 billion per year, and societal costs of up to £10 billion [10]. Guidelines could help in this clinical context.

Clinical guideline NG51 *Sepsis: recognition, diagnosis, and early management* was published by the National Institute for Health and Care Excellence (NICE) in 2016 [11]. This provides national recommendations for recognition, diagnosis, and management of sepsis in all patient populations in UK, based on a systematic review of efficacy and cost-effectiveness. It recommends considering the use of an early warning score for suspected sepsis in acute hospital settings. The National Early Warning Score (NEWS), the most widely used risk scoring system in the UK, is based on 7 physiological parameters. NEWS was developed in 2012 [12] and updated to NEWS2 in 2017 [13]. Red Flag Sepsis, a flow chart which recommends different management based on the likelihood of organ dysfunction [14], built upon NEWS by combining an easy-to-follow pathway with unambiguous recommendations.

NICE acknowledges that the application of its clinical recommendations is not mandatory, given that local and individual circumstances must also be taken into consideration. Both NICE and the Royal College of Emergency Medicine (RCEM), author of three national audits of severe sepsis and septic shock since 2011/12, recommend that all emergency departments have a sepsis lead and their own individualized sepsis protocol. The RCEM reports that this is the case in 95% and 96% of departments respectively [15]. Thus, this seems an appropriate clinical context where the variation between local guidelines can be assessed. 

In this paper, we analyze local guidelines for the identification and risk stratification of adult patients with suspected sepsis in non-specialist acute hospitals (NHS Trusts) in England. Our aim is to investigate variation in care pathways and discuss its implications. To our knowledge, this is the first formal evaluation of its kind in relation to local sepsis guidelines.

## 2. Materials and Methods

We obtained local sepsis guidelines from non-specialist acute NHS Trusts in England, and extracted information about the pathway for identification of suspected sepsis. Our scope was to include guidelines applicable to a general adult population, not those relating only to specific groups such as pregnant patients or those with suspected neutropenic sepsis.

### 2.1. Sampling Strategy

We aimed to include a representative sample of 10% of Trusts in England, which we considered to be achievable based on the resources available, and a good balance between resource and the value of the exercise. In choosing those to request, we sought to include variation in factors that may influence organizational operation, such as geographical location, size, and governance arrangements. Thus, our sampling strategy was designed to capture variation in practice while minimizing the required sample size. The choice of Trusts was arbitrary other than varying these factors, and the inclusion of our own Trust. Table 1 shows characteristics of 15 Trusts selected, which included representation from all nine regions of England. The sizes of the Trusts, as measured by the number of general and acute beds [16], ranged from the 25–30th to the 95–100th centiles. The sample included both foundation and non-foundation Trusts, and both teaching and non-teaching Trusts.

### 2.2. Acquisition of Guidelines

Guidelines were requested by freedom of information (FOI) requests in August 2018. The Freedom of Information Act 2000 gives anyone the right to request information from public bodies in the UK, who should respond within 20 working days [17]. Trusts’ FOI departments were contacted by email and asked to provide: “A copy of your current or most recent local adult sepsis protocol (i.e., guidance for staff for identification of suspected sepsis in adults), with publication dates and expiry dates where available. If such a document does not exist, a response to indicate that this is the case.”

### 2.3. Data Extraction

Responses were deemed valid for analysis if they included a guideline for the identification of sepsis in adults. Two independent reviewers (authors AB and EK) read each response and listed all physiological signs, clinical signs, and risk factors included for identification of suspected sepsis. These items were consolidated into groupings by our local sepsis lead (author BM) as shown in Appendix A. Physical signs were grouped into categories which best reflected their position on a NEWS chart. Site of infection was also divided into the anatomical site or physiological system affected. A data extraction strategy was agreed and documented as a set of instructions (Appendix B) and data extraction form (Appendix C). The strategy involved determining for each guideline the use of NEWS, then for each step in the guideline the included signs and risk factor items up to the recommendation to prescribe antibiotics. A step was deemed to be a question or criterion (or collection of these considered together), the outcome of which determined how the patient progressed in the pathway. Data were extracted from the responses independently by the same reviewers according to these instructions, with discrepancies resolved by discussion including a third reviewer (author SG) to reach agreement and arrive at a final set of reference results. This group also selected the pathways that they deemed to be the simplest and most complex.

### 2.4. Validation Exercise

To validate the reproducibility of the reference results, two independent reviewers who were not involved in the process described above (authors AW and AJ) were provided with the instructions, item groups, and guidelines to perform data extraction. To evaluate the feasibility of this approach as a standard methodology, the reviewers recorded the length of time taken for analysis of each guideline. The data extraction error rates were calculated by dividing the number of errors by the number of potential errors (considering included and excluded items, inclusion of NEWS, steps numbers, and classification of steps as contributing to an indication of high risk or lower-risk sepsis).

## 3. Results

All 15 Trusts responded to the FOI request throughout September and October 2018, with response times ranging from 3 to 39 working days. Three responses were excluded prior to data extraction: one Trust (14 in Table 1) did not have a policy covering sepsis in a general adult population (only one for neutropenic sepsis), another (12 in Table 1) returned an emergency department assessment form without a sepsis guideline, and a third (13 in Table 1) returned a policy covering the management of deteriorating adult patients without a sepsis guideline. This left 12 guidelines available for the data extraction exercise.

The remaining guideline publication dates ranged from 2012 to 2017, with the majority (9/12) having been published within the two years prior to the analysis. The guidelines ranged in length from a single page to 16 pages, and had different formats, with information presented variably as prose, lists, tables, and flow diagrams.

One further guideline (11 in Table 1) was excluded after attempting data extraction because discrepancies within it prevented step numbers and included items being determined, leaving 11 guidelines for analysis (1–10 and 15 in Table 1). Figure 1 summarizes NEWS scores, physiological signs, clinical signs, and risk factors included in each guideline for the identification of sepsis. A more detailed version of this table, showing items included by step number, is given in Appendix D. Two responses described the use of an electronic screening system, into which clinical information is entered and an alert triggered if a patient is deemed to be at high risk of having sepsis on the basis of algorithms (details of which were not included in the responses).

### 3.1. Use of Published Scoring Systems and Pathways

The aggregate NEWS score was incorporated into 7 Trusts’ guidelines, and the use of individual NEWS scores into 5. The use of NEWS in one guideline was indeterminate due to the statement “Does your patient have an EWS (Early Warning Score) ≥ 3...”. This is because an individual NEWS score cannot exceed 3, but a threshold of 3 is not normally applied to aggregate score. The relationship between EWS and NEWS was also unclear. Three Trusts (guidelines 6, 7, 8 in Figure 1) had adopted the UK Sepsis Trust template [14], but with slight variations. For example, two Trusts’ guidelines required any amber flag to be present to proceed to the next step, whereas the third required two amber flags, and two Trusts’ guidelines included lactate in a fifth step, but the third did not.

### 3.2. Physiological Signs and Thresholds

According to the groupings determined by BM (Appendix A), 15 signs and risk factor grouping items were included in the guidelines for identification of sepsis. The number of items included in each guideline for primary identification of high-risk sepsis ranged from 4 to 13 (median 11, interquartile range 2.5). Mental state changes were included in all guidelines, with respiratory rate, oxygen, blood pressure, pulse rate, and signs of infection included in all but one. The least common items were blood/lab results, appearing in 2 guidelines, and organ failure, appearing in 3.

### 3.3. Pathway Structure

Guidelines contained between 2 and 5 steps (median 4, interquartile range 2.5) up to the recommendation to prescribe antibiotics. Figure 2 presents flow diagrams, one depicting the sepsis identification pathway deemed to be the simplest, comprising 2 steps including 11 items (guideline 15 in Figure 1 and Appendix D), and the other showing that deemed to be the most complex, comprising 5 steps including 14 items (guideline 8 in Figure 1 and Appendix D).

### 3.4. Validation Exercise

The data extraction error rates, when comparing results from guideline-naïve reviewers to the reference results, were 10% and 13% for raters 1 and 2, respectively. There were 133 errors in total (rater 1 = 56, rater 2 = 77): 35 (15, 20) instances of items in the guidelines being missed, 3 (2, 1) of items being included that were not in the guideline, 88 (34, 54) cases of step numbers being misinterpreted, 6 (4, 2) cases of items being allocated to the wrong group (as defined in Appendix A), and 1 (1, 0) instance of a rater classing a non-NEWS item as included in NEWS. Both raters took a mean of 8 minutes to read each guideline, with times ranging from 1 to 23 minutes.

## 4. Discussion

### 4.1. Variation in Sepsis Guidelines

Substantial variation was observed between local guidelines in the identification of patients with suspected sepsis. The number of items used to recognize high risk patients varied between 4 and 13 items out of a possible 15, based on signs and symptoms groupings. Pathways ranged from simple, containing only 2 steps, to more complex, comprising 5 steps and including risk stratification decisions. The date of our information request was such that guidelines were unlikely to reflect the NEWS2 update 8 months prior [13], and indeed all guidelines preceded this update. Most of the guidelines we analyzed did refer to NEWS, but not all, and the seven NEWS parameters were not always included in those that did not. For example, a very low or high temperature would not feed into the process, as described in one guideline, of determining sepsis risk. We did not analyze differences arising from the number of red or amber flags required to proceed in one step within a pathway (one, two, or more), or the thresholds applied to physiological measurements.

These findings support those published previously using different methodology in this disease area and others [18,19,20,21]. Rhee et al [22] found by asking critical care specialists to classify clinical vignettes that application of the traditional consensus sepsis definition resulted in large variability. The same group later reported variation in the time taken for sepsis to be identified, and management initiated [23]. Even after training to standardize the process, there was low agreement between clinicians reviewing cases to identify these time points.

### 4.2. The Potential Impact of Uncertainty in Disease Prevalence on POCT Evaluation and Adoption

To facilitate adoption of a new POCT in the healthcare system, the performance, safety, effectiveness, cost-effectiveness and affordability of a new test has to be demonstrated compared to the current care pathway for the patient population of interest [24,25,26,27].

POCT performance, safety and effectiveness are evaluated with clinical studies and the starting point of the design of any clinical study is disease prevalence [28]. Disease prevalence is directly linked to the concept of disease definition, and an operational definition of the disease can be linked to when a patient is defined as “diseased”, i.e., in need of treatment, in the guidelines. If guidelines across hospitals are very variable, the disease definition is not clear, and its prevalence will vary across the country. This has been observed previously in sepsis [10].

Prevalence is an essential consideration during the planning of clinical studies to decide the sample size needed to observe (or not observe) the outcome of interest with reasonable certainty [29]. Using an inaccurate or wide-ranging prevalence may result in the failure of the study, or in outcome measures that are too imprecise to be useful. These difficulties have been acknowledged previously in the context of sepsis [30]. The implication is that evaluations of new POCTs in this clinical context may be delayed because of a lack of a shared operational definition of the disease, reflected in the variability of local guidelines.

Diagnostic test accuracy refers to the ability of a test to discriminate between those who have and those who do not have the target condition. Certain measures of diagnostic accuracy are intrinsically linked to disease prevalence, including the positive and negative predictive values (PPV and NPV). These are the probabilities that a patient who tests positive has the disease, or one who tests negative does not have the disease, respectively. More recently it has also been shown that sensitivity (the proportion of patients with the target condition who have a positive test result) and specificity (the proportion of those without the target condition who have a negative test result) are likely to depend on prevalence if these measures are evaluated in different clinical settings, e.g., emergency departments versus intensive care units [28]. Without a robust assessment of prevalence, the performance of the test (its clinical validity) cannot be evaluated with precision.

To assess cost-effectiveness and affordability economic evaluations are used [31,32]. Economic evaluations consist of a “map” of processes (e.g., disease diagnosed, test results, treatment) and outcomes (e.g., death or survival) and the probabilities that those processes and outcomes occur in the population of interest (e.g., prevalence, sensitivity, and specificity of the test, mortality rate). We have already discussed that sensitivity and specificity may depend on prevalence. Probabilities are essential parameters in an economic evaluation. A large uncertainty surrounding these essential probabilities makes the analysis less helpful and more difficult to interpret. This is a common problem, even when combining multiple tools to predict economic factors, as shown by Quercioli et al [33], who developed a composite measure of health to predict expenditure.

Finally, from a practical perspective, if only one (or a small number of) care pathway(s) needed to be modified on a national scale, as opposed to a large number of local pathways, a new POCT could be implemented more quickly and at less expense.

### 4.3. The Potential Impact of Variable Care Pathways in POCT Evaluation and Adoption

As mentioned in the previous section, economic evaluations are based on a model structure that maps processes and outcomes. This structure is built on the care pathway because it is an overall representation of the clinical decisions of patients, nurses and doctors [1,32]. If the point in the pathway where the test might fit is different across hospitals because their care pathways differ, then the structure of the economic model will need to be adapted (e.g., with sensitivity analysis [32]) or a more high level structure may be used. This could be particularly problematic in the assessment of affordability. This analysis is relied upon by budget holders when deciding whether to implement a new technology in their region/hospital. High variability in the care pathways increases the complexity of the economic analysis for the evaluation of the POCT during the earlier stages of the evidence generation, but also at the point of widespread adoption when affordability needs to be demonstrated to the budget holders.

The care pathway also informs the protocol of clinical studies: the point in the pathway where the test might fit defines the patient population of interest and the series of decisions in the pathway defines the study processes. A protocol based on one pathway when there is large variability across the country is unlikely to be generalizable or transferable to other settings.

Outcomes (that often represent patient health) are essential in economic evaluations but are also the endpoints of clinical studies. Outcomes are also affected by variation in the pathways. In our sepsis case study, for example, varying pathways could mean a difference in the group of patients treated with antibiotics for suspected sepsis within each organization. With each step in a pathway, patients not fulfilling the criteria are diverted, and so Trusts with longer pathways, by, for example including more signs and symptoms, are likely to treat fewer patients. In addition, the number of steps and items to consider when deciding whether to prescribe antibiotics may influence the timeliness of any intervention. Thus, variations in guidelines may affect patient outcomes [20,34]. This is likely to have contributed to the finding by Churpek et al [35] that different definitions of sepsis lead to variation in mortality rate. Conversely, oversimplified pathways may result in low specificity and the well-acknowledged harms of overprescription of antibiotics.

### 4.4. Strength and Weaknesses of the Methodology

To our knowledge, this analysis is the first of its kind, irrespective of medical specialty. Our approach uses readily available resources and avoids biases inherent in practitioner surveys [36], and those due to patient selection in clinical studies. However, the analysis does not capture differences between guidance and actual practice, or provide insight into how guidelines were developed and thus the reasons for variation [36]. FOI requests are an efficient way of obtaining information from public authorities, but the process after a request reaches the FOI contact is hidden, so the source of the information within the organization is unknown. The burden of the request to the organization should be weighed up with the benefits of the research. If all guidelines were made publicly available online (as some, including our own Trusts, are), assessment of variation in local pathways would be more easily achieved, increasing transparency of care.

Our data extraction methodology, involving two independent reviewers, aligns with that of systematic literature reviews [37]. Our validation exercise demonstrated error rates lower than the majority reported in a recent review [38]. This novel approach therefore may be robust. We did not aim to appraise guideline quality or the development process as existing instruments [36,39] were not suitable.

A study by Drennan et al [40], in which 98 local continence policies were obtained from the public domain and written requests to 38 sites, had methodological similarities with our work. Their study included a much larger sample than ours, but little information was given about their data extraction and analysis methodology. Rather, the methods section cited theoretical qualitative frameworks. The same is true of another study all eight national policy documents and guidelines on palliative sedation [41]. In addition, neither paper described the data extraction or analysis being carried out by two or more independent reviewers, as we did.

We devised a strategy to minimize bias in sampling Trusts, while considering factors that may affect the way in which Trusts operate, including a similar number of sites to that represented in several recently published, high-profile trials [42,43,44]. We demonstrated considerable variation based on a selected 10% of Trusts.

Responses from four of the five Trusts with the fewest beds were excluded; one because of discrepancies within the guideline and the others because a guideline matching the request was not received. Therefore only 12 of 15 Trusts (80%) had a local sepsis guideline, compared to 96% recently reported by the RCEM [15]. This may suggest that smaller Trusts are less likely to have a (comprehensive) local sepsis guideline for the general population. Unfortunately, this means that smaller organizations are poorly represented in our results. However, we do not believe that better representation of small Trusts would alter our conclusion that there is large variation across sepsis guidelines, which is likely to correlate with large variation in clinical practice. Indeed, the lack of a guideline is likely to cause variation in clinical practice even within, as well as between, Trusts.

## 5. Conclusions

We have proposed a novel methodology for analyzing variation in local clinical guidelines, and using this we found large variation in the identification of patients with suspected sepsis across UK hospitals.

We advise POCT developers to analyze and visualize the care pathway of patient groups that are likely to benefit from their technology at the early stages of development (using standard methods, such as interviews, or the methodology proposed in this article). This would allow them to understand if their technology has the potential for integration into the current pathway and to be useful to patients, clinicians, and/or laboratory technicians. As we have shown in this discussion, a possible barrier to adoption of a new technology is the variability of the care pathways. It is useful to identify this issue in the early stages of POCT evaluations so that it can be taken into account when planning and conducting economic and clinical studies, and the route to widespread adoption.

## Figures and Tables

**Figure 1 ijerph-17-02026-f001:**
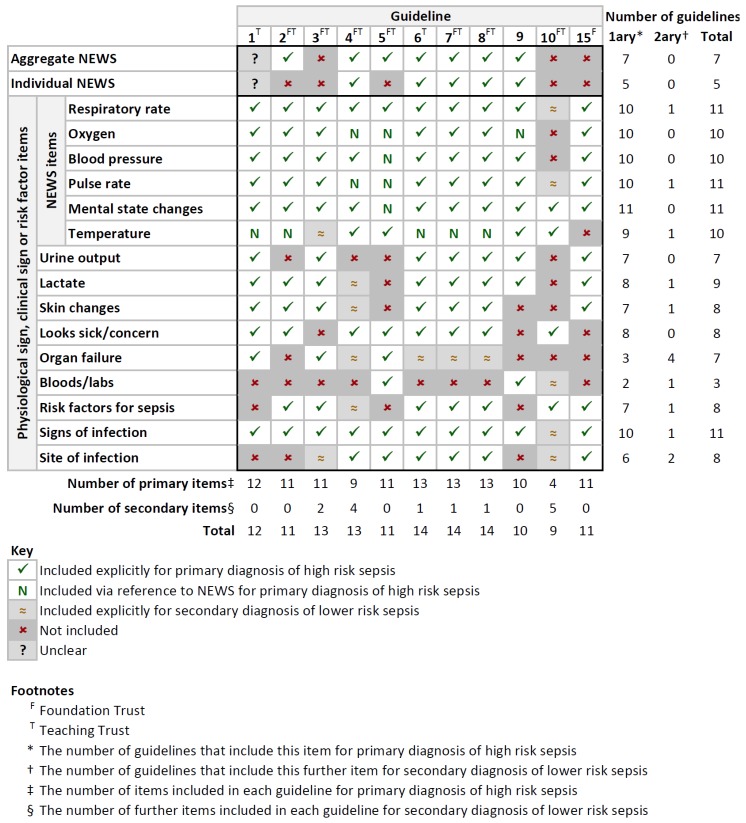
Reference results: A summary of NEWS scores, physiological signs, clinical signs, and risk factors included in each guideline for the identification of sepsis. Trusts are ordered from left to right by descending number of general and acute beds [16].

**Figure 2 ijerph-17-02026-f002:**
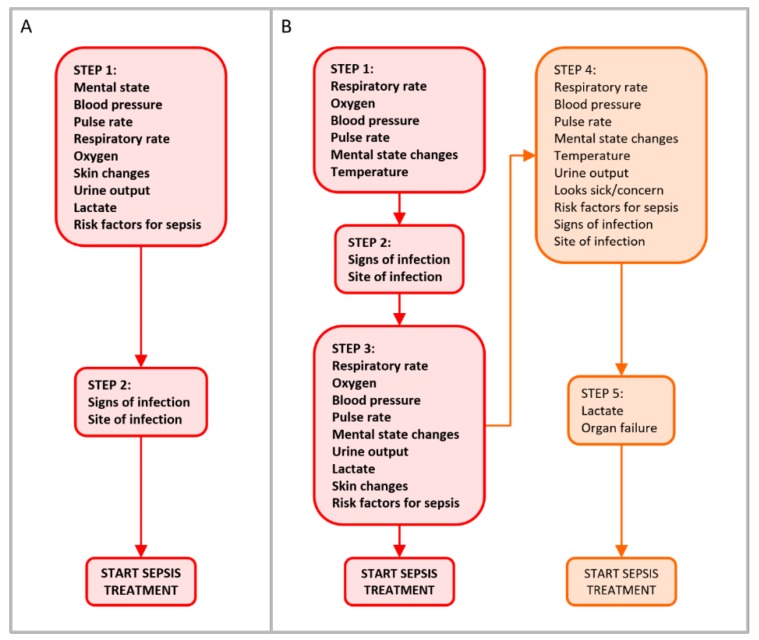
Diagrams depicting two sepsis identification pathways, the simplest (A) comprising two steps including 11 items (guideline 15 in Figure 1 and Appendix D), and the most complex (B), comprising five steps including 14 items (guideline 8 in Figure 1 and Appendix D). Steps shown in bold font relate to primary identification of high-risk sepsis, and those in non-bold font (those on the right in B) relate to secondary diagnosis of lower-risk sepsis.

**Table 1 ijerph-17-02026-t001:** Characteristics of the 15 NHS England Trusts whose local sepsis guidelines were requested, ordered by descending size as measured by the number of general and acute beds as a centile range [16].

#	Region of England	Size	Foundation?	Teaching?	Reason of Exclusion
1	East Midlands	95–100%	No	Yes	
2	North-East	95–100%	Yes	Yes	
3	South-East	85–90%	Yes	Yes	
4	Yorkshire and the Humber	80–85%	Yes	Yes	
5	London	75–80%	Yes	Yes	
6	London	70–50%	No	Yes	
7	North-West	70–50%	Yes	Yes	
8	South-West	65–70%	Yes	Yes	
9	South-West	60–65%	No	No	
10	North-West	55–60%	Yes	Yes	
11	North-West	45–50%	No	Yes	Discrepancies in guideline
12	North-East	40–45%	Yes	No	Response did not include a sepsis guideline
13	West Midlands	40–45%	No	No	Response did not include a sepsis guideline
14	East of England	35–40%	Yes	Yes	Response did not include a sepsis guideline for a general adult population
15	Yorkshire and the Humber	25–30%	Yes	No

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
