# Peer review of "High Variability in Sepsis Guidelines in UK: Why Does It Matter?"

_ijerph, 2020, doi:10.3390/ijerph17062026_

Round 1
Reviewer 1 Report
This is an interesting study about a new methodology aimed to evaluate multiple guidelines in a faster and more reliable way, aligning with that of systematic reviews. The topic has huge implications in the outcomes, prevalence and economic factors of many disease, as the Authors describe in a comprehensive and interesting way.
I would suggest some changes and corrections:
- Lines 89-97, “Sampling Strategy” – I would add some more details about the selection criteria of the Trusts that eventually were requested. Was any random factor involved?
- Line 90 – Was there a particular reason behind the choice of the 10% size for the sample of Trusts in England?
- Line 111 – I would add more details about the item grouping criteria that were chosen and used in the case of sepsis, as presented in Appendix A, especially about the distinction between Signs and Site of Infection. Was that the case for the errors of item being allocated to the wrong group? (Line 190)
- Table 1 – Trusts from 6 to 10 are missing. Also, the superscript “1”, to indicate the trusts’ guidelines not included in the analysis, are not showed. Should that concern the four of the five Trusts with the fewest beds? (Lines 306-309)
- Figure 1 – Although the Authors have provided this information in Table 1, I would add whether the Trust were a Teaching Hospital or not also in this Figure
- Line 246 – I would add that this is a common problem, even when combining more than one tool to predict economic factors, as in the following example:
- Quercioli C, Nisticò F, Troiano G, Maccari M, Messina G, Barducci M, et al. Developing a new predictor of health expenditure: preliminary results from a primary healthcare setting. Public Health. 2018 Oct;163:121–7.
- Lines 304-305 – I’m not absolutely certain if it’s totally correct to say that a larger sample size would be unlikely to change the conclusions, especially since small hospitals were poorly represented, as the Authors state later. Maybe it would be unlikely just for larger hospitals?
Author Response
We wish to thank the reviewer for the time spent reading our work and suggestions to improve it.
Lines 89-97, “Sampling Strategy” – I would add some more details about the selection criteria of the Trusts that eventually were requested. Was any random factor involved?
We have added further detail under the Sampling strategy heading within Materials and Methods.
Line 90 – Was there a particular reason behind the choice of the 10% size for the sample of Trusts in England?
We have added further detail under the Sampling strategy heading within Materials and Methods.
Line 111 – I would add more details about the item grouping criteria that were chosen and used in the case of sepsis, as presented in Appendix A, especially about the distinction between Signs and Site of Infection. Was that the case for the errors of item being allocated to the wrong group? (Line 190)
We have added further detail under the Data extraction heading within Materials and Methods. None of the rater errors were due to misallocation between the ‘signs of infection’ and ‘site of infection’ groups so we have not added this to the manuscript.
Table 1 – Trusts from 6 to 10 are missing. Also, the superscript “1”, to indicate the trusts’ guidelines not included in the analysis, are not showed. Should that concern the four of the five Trusts with the fewest beds? (Lines 306-309)
Trusts 6 to 10 were accidently omitted from Table 1 and have been added. We have also added a column to indicate why Trusts were excluded.
Figure 1 – Although the Authors have provided this information in Table 1, I would add whether the Trust were a Teaching Hospital or not also in this Figure
We have added footnotes to Figure 1 and Appendix C to indicate which Trusts are Teaching and Foundation Trusts.
Line 246 – I would add that this is a common problem, even when combining more than one tool to predict economic factors, as in the following example: Quercioli C, Nisticò F, Troiano G, Maccari M, Messina G, Barducci M, et al. Developing a new predictor of health expenditure: preliminary results from a primary healthcare setting. Public Health. 2018 Oct;163:121–7.
We have added this reference to the second-last paragraph under the The potential impact of uncertainty in disease prevalence on POCT evaluation and adoption heading within Discussion.
Lines 304-305 – I’m not absolutely certain if it’s totally correct to say that a larger sample size would be unlikely to change the conclusions, especially since small hospitals were poorly represented, as the Authors state later. Maybe it would be unlikely just for larger hospitals?
We have reworded the last two paragraphs within Discussion to clarify our view that better representation of smaller Trusts would be unlikely to reduce the amount of variation found.
Reviewer 2 Report
The Authors present the methodology which extrapolates the care pathway from local hospital guidelines and assesses their variation in case of sepsis. They describe in details sampling strategy, acquisition of guidelines, data extraction and validation exercise. They focuse on possible application of this methodology for POCT evaluation and adoption.
However, I would suggest to list/ give an example of POCT which currently apply for sepsis and examples of the tests which potentially coud have such application. In my opinion it would be interesting for readers.
Author Response
We wish to thank the reviewer for the time spent reading our work and suggestions to improve it.
I would suggest to list/give an example of POCT which currently apply for sepsis and examples of the tests which potentially could have such application. In my opinion it would be interesting for readers.
We have added reference to procalcitonin testing in the introduction.